# Dual-mode recognition of tRNA^Pro isoacceptors by *Toxoplasma gondii* Prolyl-tRNA synthetase

Indira Rizqita Ivanesthi[1,4], Emi Latifah[1,4], Shih-Yang Liu[1], Yi-Kuan Tseng[2], Hung-Chuan Pan[3] & Chien-Chia Wang 🆔 [1✉]

## Abstract

Prolyl-tRNA synthetases (ProRSs) exhibit diverse domain architectures and motifs, evolving into prokaryotic (P-type) and eukaryotic/archaeal (E-type) variants. Both types exhibit high specificity for the recognition and aminoacylation of their cognate tRNAs. Interestingly, the parasitic eukaryote *Toxoplasma gondii* encodes a single E-type ProRS (TgProRS) but utilizes two distinct tRNA^Pro isoacceptors: a cytosolic E-type (with C72/C73) and an apicoplast P-type (with G72/A73). Our study demonstrates that TgProRS, despite being classified as an E-type enzyme, efficiently charges both tRNA^Pro isoacceptors and functionally compensates for yeast cytoplasmic and mitochondrial ProRS activities. Notably, while C72/C73 are dispensable for cytosolic tRNA^Pro charging, G72/A73 are crucial for apicoplast tRNA^Pro aminoacylation. Furthermore, Mutations in the motif 2 loop selectively affect E- or P-type tRNA^Pro recognition. While TgProRS exhibits similar susceptibility to azetidine (a proline mimic) when charging both tRNA^Pro types, cytosolic tRNA^Pro charging is five times more sensitive to inhibition by halofuginone (a Pro-A76 mimic) compared to apicoplast tRNA^Pro charging. These findings underscore TgProRS's dual functionality, showcasing its remarkable evolutionary adaptability and providing valuable insights for developing more selective therapeutic agents.

**Keywords** Aminoacyl-tRNA Synthetase; Halofuginone; Parasite; Protein Synthesis; Toxoplasmosis
**Subject Categories** Microbiology, Virology & Host Pathogen Interaction; RNA Biology; Translation & Protein Quality

## Introduction

Aminoacyl-tRNA synthetases (aaRSs) are essential enzymes that catalyze the attachment of amino acids to their corresponding tRNAs, forming aminoacyl-tRNAs. These charged tRNAs are subsequently delivered to ribosomes, where they decode the genetic code by pairing anticodons with mRNA codons to produce proteins

(Carter, 1993). Therefore, the precise aminoacylation of tRNA by aaRSs is crucial for maintaining the fidelity of translation. Each aaRS specifically recognizes its corresponding tRNA by identifying unique identity elements within the tRNA structure. These elements, typically located in the acceptor stem and anticodon, are usually conserved among tRNAs charged with the same amino acid but are absent in other tRNAs (Schimmel et al, 1993).

There are at least 20 aaRSs, divided into two classes: class I and class II. Class I has a Rossmann fold with conserved motifs (HIGH and KMSKS), while class II features a seven-stranded β-sheet surrounded by α-helices and three less rigid motifs (I, II, and III). The two classes differ in enzyme kinetics, structure, size, oligomeric state, and sequence similarity (Carter, 1993; Eriani et al, 1990; Kwon et al, 2019). Prolyl-tRNA synthetase (ProRS) belongs to class II aaRS that catalyzes the attachment of proline to tRNA^Pro (Bartholow et al, 2014). ProRSs exhibit two distinct domain architectures: E-type (eukaryote/archaeon-like) and P-type (prokaryote-like). E-type ProRS includes a C-terminal zinc-binding extension, while P-type ProRS features a ~180-residue insertion domain between motifs 2 and 3, essential for editing (Fig. 1A) (Beuning and Musier-Forsyth, 2001; Yaremchuk et al, 2000). Most E-type ProRSs also possess a conserved C-terminal tyrosine (Y) residue that stabilizes the aminoacylation active site (Fig. 1A) (Yaremchuk et al, 2001). Additionally, yeast ProRS has a unique N-terminal ProXp-ala domain, and in metazoans, ProRS fuses with GluRS to form glutamyl-prolyl-tRNA synthetase (EPRS), a bifunctional synthetase (Fig. 1A) (Kim and Kang, 2022; SternJohn et al, 2007). Beyond protein synthesis, EPRS plays noncanonical roles, including regulating inflammatory responses via the GAIT complex and promoting proliferation in estrogen receptor-positive breast cancer, where high expression correlates with tamoxifen resistance and poor prognosis, making it a potential therapeutic target (Arif et al, 2009; Katsyv et al, 2016).

Previous studies have shown that E- and P-type ProRSs exhibit strict specificity for their respective tRNAs^Pro and cannot cross-acylate each other's tRNAs^Pro. In *E. coli*, major recognition elements lie in the acceptor stem (G72 and A73) as well as G35 and G36 in the anticodon loop (Liu et al, 1995; Shimizu et al, 1992). In contrast, human ProRS bypasses the corresponding C72 and C73 in the acceptor stem, instead depending primarily on G35 and G36 in the anticodon for tRNA^Pro recognition (Burke et al, 2001;

[1]Department of Life Sciences, National Central University, Jungli District, Taoyuan 32001, Taiwan. [2]Graduate Institute of Statistics, National Central University, Jungli District, Taoyuan 32001, Taiwan. [3]Department of Neurosurgery, Taichung Veterans General Hospital, Taichung 407219, Taiwan. [4]These authors contributed equally: Indira Rizqita Ivanesthi, Emi Latifah. ✉E-mail: dukewang@cc.ncu.edu.tw

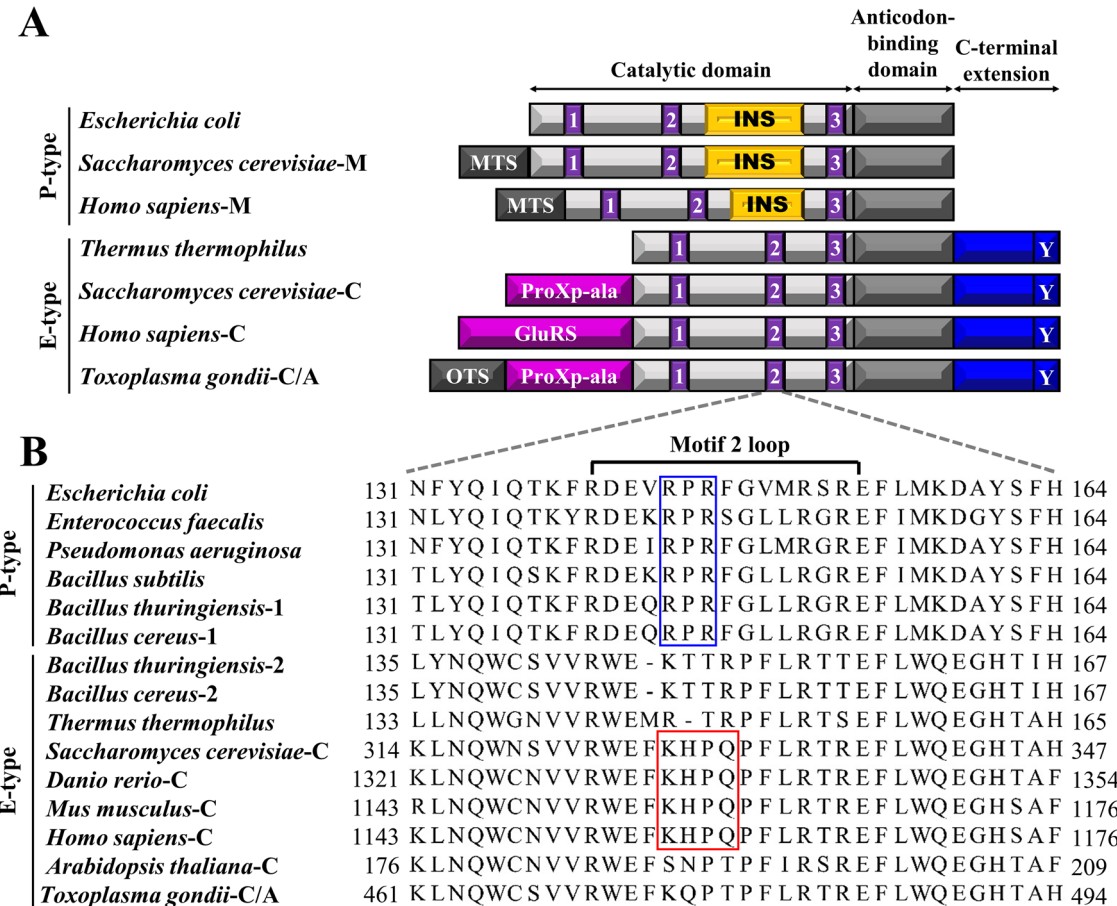

**Figure 1. Structural organization of E- and P-type ProRSs.**

(A) Domain organization of ProRS. INS insertion domain, MTS mitochondrial targeting signal, OTS organellar targeting signal, A apicoplast, C cytosol, M mitochondrial. TgProRS is co-localized in the cytosol and apicoplast. (B) Alignment of motif 2 sequences in ProRSs. Motif 2 sequences were aligned from bacterial and eukaryotic cytoplasmic and organellar ProRSs. The conserved RPR motif in bacterial P-type ProRSs is highlighted with a blue box, while the conserved KHPQ motif in eukaryotic E-type ProRSs is highlighted with a red box.

Stehlin et al, 1998). Notably, simply swapping C72/C73 with G72/A73, or vice versa, is insufficient for *E. coli* or human ProRS to aminoacylate the mutated non-cognate tRNA$^{Pro}$. An additional swap of the anticodon-D stem biloop is required for ProRS to charge the mutated non-cognate tRNA$^{Pro}$ (Burke et al, 2000; Stehlin et al, 1998). In addition to the typical recognition mechanisms of E- and P-type ProRSs, previous studies estimated that ~22% of bacteria possess an E-type ProRS (Vargas-Rodriguez and Musier-Forsyth, 2013), which has been shown to specifically charge P-type tRNA$^{Pro}$ (Ivanesthi et al, 2024).

The phylum Apicomplexa includes intracellular protozoan parasites, many of which pose significant threats to human and animal health. *Toxoplasma gondii* is one such eukaryote, capable of causing spontaneous abortion, congenital birth defects, and severe illness or death in warm-blooded animals (Manickam et al, 2022; Pino et al, 2010). Apicomplexans have three translational compartments: the cytosol, a single tubular mitochondrion, and the apicoplast. The mitochondrial genome of *T. gondii* is highly reduced, with only three open reading frames (ORFs), and lacks tRNA genes, relying instead on the import of charged cytosolic tRNAs (Esseiva et al, 2004). Although all compartments require a complete set of charged tRNAs, the apicomplexan nuclear genome lacks enough aaRS genes for compartment-specific targeting. Studies have shown that *T. gondii*'s aaRSs are either cytosolic, apicoplast-targeted, or shared between the two compartments, but absent from the mitochondrion (Pino et al, 2010).

Sequence analysis reveals that alternative translation initiation of the sole ProRS gene in *Toxoplasma gondii* generates two isoforms: a longer form (initiated at AUG1), which includes an organellar targeting signal (OTS) (residues 1–119) for apicoplast localization, and a shorter form (initiated at AUG120), which is retained in the cytoplasm (Fig. 1A). *T. gondii* ProRS (TgProRS) displays features of E-type ProRS, including a C-terminal extension domain and the absence of an insertion domain (Fig. 1B). Interestingly, this eukaryote encodes two distinct tRNA$^{Pro}$ iso-acceptors: a cytosolic E-type tRNA$^{Pro}$ and an apicoplast P-type tRNA$^{Pro}$ (Fig. 2A). This prompted us to explore whether TgProRS can charge distinct tRNA$^{Pro}$ isoacceptors and uncover the mechanisms behind this ability. Our findings reveal that TgProRS is an enzyme with relaxed specificity that employs distinct mechanisms to recognize both its cognate and otherwise non-cognate tRNA isoacceptors.

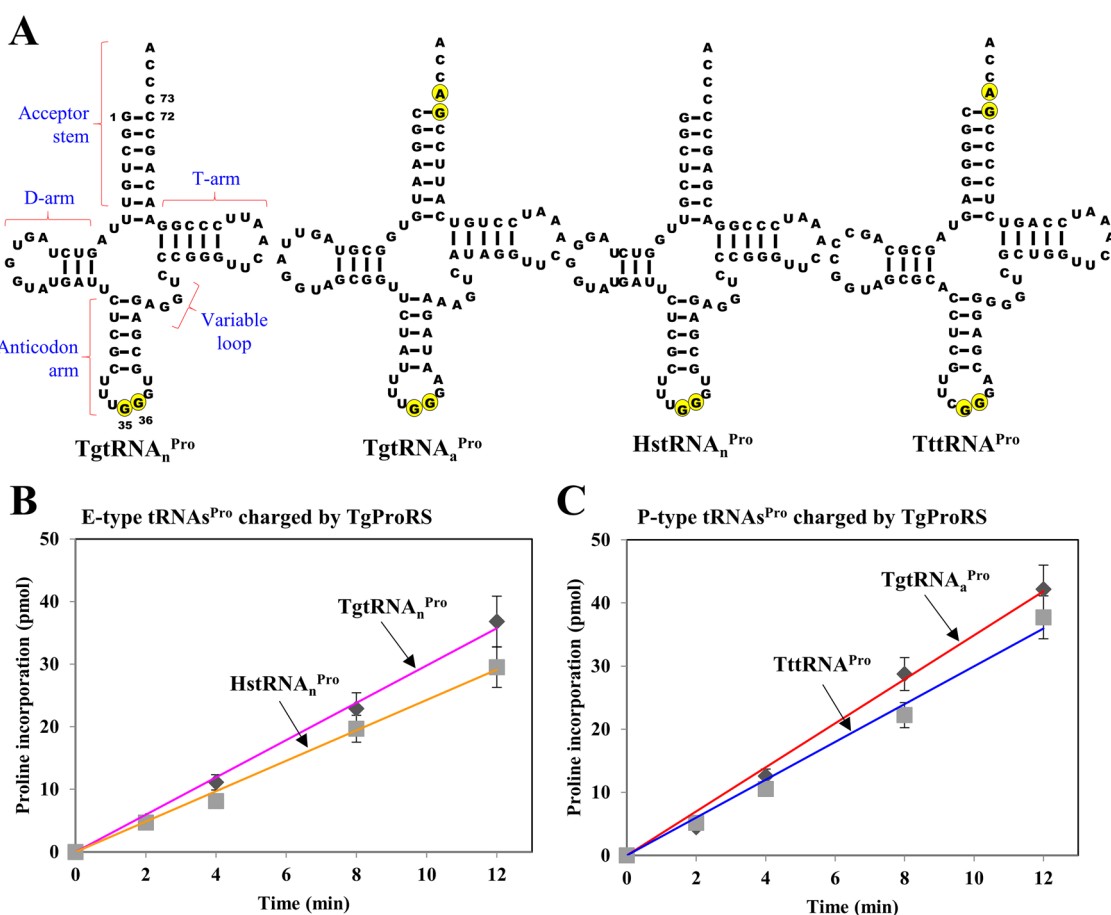

**Figure 2. Aminoacylation of E- and P-type tRNA$^{Pro}$ isoacceptors by TgProRS.**

(A) Cloverleaf structure of tRNA$^{Pro}$. Primary identity elements of tRNA$^{Pro}$ are highlighted in yellow. TgtRNA$_n^{Pro}$, *T. gondii* nuclear-encoded cytoplasmic tRNA$^{Pro}$; TgtRNA$_a^{Pro}$, *T. gondii* apicoplast-encoded apicoplast tRNA$^{Pro}$; HstRNA$_n^{Pro}$, *H. sapiens* nuclear-encoded cytoplasmic tRNA$^{Pro}$; TttRNA$^{Pro}$, *T. thermophilus* tRNA$^{Pro}$. (B) Aminoacylation of E-type tRNAs$^{Pro}$ by TgProRS. (C) Aminoacylation of P-type tRNAs$^{Pro}$ by TgProRS. Aminoacylation by TgProRS (100 nM) was carried out at 37 °C. Error bars represent ± SD; (*n* = 3 technical replicates). Source data are available online for this figure.

# Results and discussion

## TgProRS is an enzyme with relaxed specificity

To evaluate the aminoacylation activity and specificity of TgProRS, the enzyme—devoid of its OTS and ProXp-ala domains—was purified to near homogeneity using Ni-NTA column chromatography. It was then tested using in vitro-transcribed E- and P-type tRNAs$^{Pro}$ as substrates. In addition to its cognate tRNAs, TgtRNA$_n^{Pro}$ (E-type) and TgtRNA$_a^{Pro}$ (P-type), we selected *Homo sapiens* tRNA$_n^{Pro}$ (HstRNA$_n^{Pro}$), an E-type tRNA$^{Pro}$ with C72 and C73, and *Thermus thermophilus* tRNA$^{Pro}$ (TttRNA$^{Pro}$), a P-type tRNA$^{Pro}$ with G72 and A73 (Fig. 2A), as representative substrates due to their well-documented roles in ProRS aminoacylation studies (Latifah et al, 2024; Stehlin et al, 1998). The reactions were performed at 37 °C, the temperature shown in previous studies to be optimal for TgProRS activity (Jain et al, 2017). This experimental setup enabled a detailed analysis of TgProRS substrate specificity across different tRNA$^{Pro}$ isoacceptors.

As shown in Fig. 2B, TgProRS efficiently aminoacylated both E-type tRNA$^{Pro}$ substrates, TgtRNA$_n^{Pro}$ and HstRNA$_n^{Pro}$, achieving nearly identical charging levels. Surprisingly, the enzyme also exhibited comparable efficiency in aminoacylating both P-type tRNA$^{Pro}$ substrates, TgtRNA$_a^{Pro}$ and TttRNA$^{Pro}$ (Fig. 2C). This unexpected finding indicates that TgProRS, despite being classified as an E-type ProRS, exhibits remarkable flexibility in substrate specificity, effectively recognizing and charging both cognate and non-cognate E- and P-type tRNA$^{Pro}$ substrates. These findings mark the first documented case of a ProRS enzyme exhibiting such broad specificity, breaking the long-held paradigm that ProRS enzymes are restricted to charging either E- or P-type tRNA$^{Pro}$ exclusively.

Notably, certain bacteria possess E-type ProRS enzymes with exclusive specificity for P-type tRNA$^{Pro}$, as seen in *T. thermophilus* and *Bacillus thuringiensis* (Ivanesthi et al, 2024; Latifah et al, 2024). Beyond ProRS, a similar adaptive mechanism is evident in mammalian mitochondrial seryl-tRNA synthetase, which employs a dual recognition strategy to aminoacylate two structurally distinct noncanonical tRNA$^{Ser}$ isoacceptors, tRNA$^{Ser}$(UGA) and tRNA$^{Ser}$(GCU). These two mitochondrial tRNAs lack the elongated variable arm, a key recognition feature of typical tRNA$^{Ser}$ (Chimnaronk et al, 2005; Shimada et al, 2001). In addition, human lysyl-tRNA synthetase (LysRS), encoded by a single nuclear gene,

has evolved to recognize both mitochondrial and cytoplasmic tRNAs[Lys] despite their distinct bacterial and eukaryotic identity elements (Shiba et al, 1997).

## TgProRS recognize different elements in E- and P-type tRNA[Pro] isoacceptors

Previous studies have shown that P-type tRNA[Pro] recognition relies on G72 and A73 in the acceptor stem and G35 and G36 in the anticodon (Liu et al, 1995), whereas E-type tRNA[Pro] recognition is mainly determined by the anticodon (Stehlin et al, 1998). Given TgProRS's dual tRNA specificity, we hypothesized that the nucleotides at positions 72 and 73 in the acceptor stem of tRNA[Pro] play a minimal role in aminoacylation. To evaluate this, we generated tRNA mutants by swapping the bases at these positions and assessed their aminoacylation efficiency. In our assays, TgtRNA$_a$[Pro] ΔC1 was used to achieve high yields of tRNA transcripts. Previous research showed that deletion of C1 from tRNA[Pro] has only minimal effects on aminoacylation (Ivanesthi et al, 2024; Liu et al, 1995).

As anticipated, substituting C72/C73 with G72/A73 in TgtRNA$_n$[Pro] resulted in only a modest 3.2-fold reduction in $k_{cat}/K_M$ values, indicating that the nucleotides at positions 72 and 73 are not the primary specificity determinants for TgProRS when charging TgtRNA$_n$[Pro] (Table 1). In stark contrast, replacing G72/A73 with C72/C73 in TgtRNA$_a$[Pro] led to a dramatic 140-fold decrease in aminoacylation efficiency. To support these conclusions, base substitutions were introduced at G72 and A73. The mutation at G72 reduced aminoacylation efficiency by 23- to 70-fold, with G72C having the strongest effect. Similarly, the mutation at A73 reduced aminoacylation efficiency by 31- to 140-fold, with A73C having the strongest effect. These findings further support

that G72 and A73 are crucial for aminoacylation of TgtRNA$_a$[Pro], underscoring the critical importance of these acceptor stem bases in the recognition of P-type tRNA[Pro]. Moreover, these results suggest that TgProRS employs distinct recognition mechanisms depending on the tRNA[Pro] isoacceptor it charges.

Typically, E- and P-type ProRSs exhibit strict specificity for their respective tRNAs[Pro], employing distinct recognition mechanisms (Burke et al, 2000; Stehlin et al, 1998). Remarkably, TgProRS efficiently aminoacylates both E- and P-type tRNAs[Pro] by recognizing distinct identity elements. While C72 and C73 are dispensable for TgtRNA$_n$[Pro] recognition, the corresponding G72 and A73 are critical for TgtRNA$_a$[Pro] charging (Table 1), indicating distinct geometric interactions with each isoacceptor. Comparison of the nucleotide sequence and cloverleaf structure of TgtRNA$_a$[Pro] with other P-type tRNA[Pro] from bacteria and organelles revealed no significant differences (Fig. EV1), suggesting that TgtRNA$_a$[Pro] has not evolved unique sequence or structural modifications specifically recognized by TgProRS.

This study highlights TgProRS's essential role in protein synthesis across the parasite's three major compartments: the cytosol, mitochondrion, and apicoplast. Inhibiting TgProRS activity disrupts protein synthesis in these compartments, impairing cellular function and ultimately compromising the parasite's survival.

## The motif 2 loop plays a critical role in the relaxed specificity of TgProRS

The motif 2 loop in many class II synthetases plays a critical role in recognizing the tRNA acceptor stem. In bacterial P-type ProRS, the conserved RPR motif—particularly the first R residue (R144)—is essential for efficient tRNA[Pro] aminoacylation. Mutation of this residue results in a >1000-fold reduction in $k_{cat}/K_M$ without affecting amino acid activation (Burke et al, 2000). Conversely, the equivalent residue in human cytoplasmic ProRS, K1084, has a minimal effect; its mutation reduces aminoacylation efficiency by less than twofold (Burke et al, 2000), underscoring a functional divergence between bacterial and human ProRSs.

Our sequence alignment uncovered a distinctive KQPT sequence in TgProRS's motif 2 loop, replacing the conserved eukaryotic KHPQ motif (Fig. 1B). Given TgProRS's dual functionality (Table 1), we hypothesize that the unique KQPT sequence has been functionally repurposed to play a critical role in tRNA[Pro] recognition. To test this hypothesis, we conducted mutagenesis on the amino acid residues within this motif.

The four targeted amino acid residues (K474, Q475, P476, and T477) within this motif were individually mutated to alanine, and the resulting proteins were assessed for aminoacylation efficiency with TgtRNA$_n$[Pro] and TgtRNA$_a$[Pro]. The K474A mutation caused a 9.5-fold reduction in aminoacylation efficiency ($k_{cat}/K_M$) for TgtRNA$_n$[Pro] but only a 2-fold reduction for TgtRNA$_a$[Pro] (Table 2), highlighting K474's crucial role in E-type tRNA[Pro] recognition. Given that mutations at C72/C73 minimally affect TgtRNA$_n$[Pro] aminoacylation, this suggests that K474 exerts its effect through mechanisms other than direct interaction with these two bases. In contrast, Q475A and T477A mutations primarily impaired the charging of P-type tRNA[Pro], reducing efficiency by 9- and 52-fold, respectively, while only slightly affecting TgtRNA$_n$[Pro] aminoacylation (1.6- and 2-fold reductions, respectively) (Table 2). These

**Table 1. Kinetic parameters for aminoacylation of TgtRNA[Pro] variants by TgProRS.**

| TgProRS | | | | |
|---|---|---|---|---|
| TgtRNA$_n$[Pro] | $k_{cat}$ (×10⁻³ s⁻¹) | $K_M$ (µM) | $k_{cat}/K_M$ (×10⁻³ µM⁻¹ s⁻¹) | Loss of specificity (x-fold) |
| WT | 40 ± 5.1 | 2.5 ± 0.6 | 16 ± 3.2 | 1 |
| C72/C73 → G72/A73 | 98 ± 20 | 21 ± 3.5 | 5.0 ± 0.2 | 3.2 |
| **TgProRS** | | | | |
| TgtRNA$_a$[Pro] | $k_{cat}$ (×10⁻³ s⁻¹) | $K_M$ (µM) | $k_{cat}/K_M$ (×10⁻³ µM⁻¹ s⁻¹) | Loss of specificity (x-fold) |
| ΔC1 | 92 ± 18 | 3.3 ± 0.4 | 28 ± 4.4 | 1 |
| ΔC1-G72/A73 → ΔC1-C72/C73 | 2.0 ± 0.5 | 8.7 ± 0.8 | 0.2 ± 0.03 | 140 |
| ΔC1-G72A | 11 ± 3.4 | 9.0 ± 1.8 | 1.2 ± 0.4 | 23 |
| ΔC1-G72C | 1.5 ± 0.3 | 4.1 ± 1.0 | 0.4 ± 0.05 | 70 |
| ΔC1-G72U | 4.5 ± 1.0 | 9.7 ± 1.8 | 0.5 ± 0.05 | 56 |
| ΔC1-A73G | 1.4 ± 0.3 | 3.9 ± 0.9 | 0.4 ± 0.04 | 70 |
| ΔC1-A73C | 3.0 ± 0.6 | 13.5 ± 2.9 | 0.2 ± 0.02 | 140 |
| ΔC1-A73U | 12.5 ± 3.6 | 14.6 ± 3.5 | 0.9 ± 0.06 | 31 |

**Table 2. Kinetic parameters for aminoacylation of TgtRNAs^Pro by TgProRS variants.**

| TgtRNA$_n$^Pro | | | | |
|---|---|---|---|---|
| TgProRS | $k_{cat}$ ($\times 10^{-3}$ s$^{-1}$) | $K_M$ (µM) | $k_{cat}/K_M$ ($\times 10^{-3}$ µM$^{-1}$ s$^{-1}$) | Loss of specificity ($x$-fold) |
| WT | 41 ± 4.8 | 2.2 ± 0.5 | 19 ± 3.2 | 1 |
| K474A | 6.0 ± 0.2 | 3.6 ± 0.2 | 2.0 ± 0.5 | 9.5 |
| Q475A | 44 ± 5.2 | 4.2 ± 1.0 | 10 ± 2.2 | 1.6 |
| P476A | 28 ± 2.2 | 2.0 ± 0.3 | 14 ± 2.8 | 1.3 |
| T477A | 20 ± 3.0 | 2.4 ± 0.2 | 8.3 ± 1.5 | 2 |
| TgtRNA$_a$^Pro | | | | |
| TgProRS | $k_{cat}$ ($\times 10^{-3}$ s$^{-1}$) | $K_M$ (µM) | $k_{cat}/K_M$ ($\times 10^{-3}$ µM$^{-1}$ s$^{-1}$) | Loss of specificity ($x$-fold) |
| WT | 100 ± 21 | 3.6 ± 0.6 | 31 ± 5.2 | 1 |
| K474A | 72 ± 12 | 4.5 ± 1.0 | 16 ± 3.4 | 2 |
| Q475A | 76 ± 15 | 21 ± 4.5 | 3.6 ± 0.9 | 9 |
| P476A | 40 ± 5.0 | 3.1 ± 0.5 | 13 ± 3.0 | 2.4 |
| T477A | 14 ± 3.4 | 23 ± 4.6 | 0.6 ± 0.1 | 52 |

findings underscore the critical role of Q475 and T477 in P-type tRNA^Pro recognition. Meanwhile, the P476A mutation had no significant effect on the aminoacylation of either tRNA^Pro isoacceptor. Together, these results reveal that TgProRS employs distinct motif 2 loop residues to differentially recognize and charge E- and P-type tRNAs^Pro.

While the KHPQ motif in human ProRS has a minimal role in tRNA^Pro recognition (Burke et al, 2000), the KQPT motif in TgProRS plays a pivotal role in recognizing both E- and P-type tRNAs^Pro. Interestingly, the KQPT motif is also present in the motif 2 loop of the cytoplasmic ProRS in *Drosophila melanogaster* (DmProRS$_c$) (Burke et al, 2001) and *Plasmodium falciparum* (PfProRS$_c$) (Jain et al, 2014). In both species, an additional P-type ProRS is responsible for charging tRNA^Pro in their organelles. Therefore, it is worth investigating whether DmProRS$_c$ and PfProRS$_c$ exhibit the same relaxed specificity.

## TgProRS can functionally substitute for both yeast *PROS1* and *PROS2*

To validate our in vitro findings, we performed an in vivo complementation assay using yeast *PROS1* and *PROS2* knockout (KO) strains, following a previously established strategy (Ivanesthi et al, 2024). The yeast nuclear genes *PROS1* and *PROS2* encode the cytoplasmic E-type ProRS (ScProRS$_c$) and mitochondrial P-type ProRS (ScProRS$_m$), respectively (Fig. 1A). The cytoplasmic tRNA^Pro (SctRNA$_n$^Pro), encoded by the nuclear genome, is an E-type tRNA^Pro, while the mitochondrial tRNA^Pro (SctRNA$_m$^Pro), encoded by the mitochondrial genome, is a P-type tRNA^Pro (Fig. 3A). ScProRS$_c$ rescued the growth defect of the *PROS1* KO strain but not the *PROS2* KO strain, even with an MTS. Conversely, ScProRS$_m$ rescued the *PROS2* KO strain but not the *PROS1* KO strain, even without an MTS (Fig. 3B).

Consistent with our in vitro findings, TgProRS successfully complemented the *PROS1* KO strain, demonstrating its ability to

charge SctRNA$_n$^Pro. When fused with an MTS, TgProRS was imported into the mitochondria, rescuing the *PROS2* KO strain by aminoacylating SctRNA$_m$^Pro. These results establish TgProRS as a relaxed specificity enzyme capable of charging both E- and P-type tRNAs^Pro. Western blot analysis verified the proper expression of all constructs, eliminating expression differences as the cause of the observed phenotypes (Fig. 3C).

We also conducted complementation assays with TgProRS mutants (Fig. 4A–C). Consistent with our in vitro findings, the K474A mutation impaired rescue of the *PROS1* KO strain but not the *PROS2* KO strain. Conversely, the Q475A and T477A mutations failed to rescue the *PROS2* KO strain but successfully complemented the *PROS1* KO strain. The P476A mutation had minimal impact on the complementation of either KO strain. Western blot analysis confirmed proper construct expression in both KO strains (Fig. 4B–D).

These findings demonstrate that the KQPT motif in TgProRS plays a pivotal role in recognizing both E- and P-type tRNA^Pro isoacceptors. Mutation of K474 selectively reduced aminoacylation of E-type tRNA^Pro, while mutations at Q475 or T477 specifically impaired the aminoacylation efficiency of P-type tRNA^Pro (Table 2; Fig. 4). As a result, a single amino acid substitution in the motif 2 loop can convert this relaxed specificity enzyme into a single-specificity ProRS for either E- or P-type tRNA^Pro. These results underscore the importance of individual motif 2 loop residues in mediating isoacceptor-specific recognition. Notably, a previous study reported that the T477A mutation leads to slower growth in *T. gondii* (Yogavel et al, 2023).

## TgProRS exhibits differential sensitivity to HF when charging E- and P-type tRNA^Pro isoacceptors

Azetidine-2-carboxylic acid (A2C) is a non-proteinogenic amino acid and allelopathic compound primarily produced by sugar beets and lilies. Due to its structural similarity to proline, A2C can bind to ProRS active sites, leading to its misincorporation into proteins. This misincorporation disrupts proper protein folding and function, ultimately impairing cellular processes (Song et al, 2017). As a result, A2C exhibits broad-spectrum inhibitory effects on various organisms, including bacteria, fungi, animals, and plants. The mechanism of inhibition is believed to stem from the replacement of L-proline with A2C in newly synthesized proteins, causing structural and functional defects (Lee et al, 2016).

Halofuginone (HF), a synthetic analog of febrifugine (FF) and a Pro-A76 mimic, is among the most potent antimalarial agents (Tye et al, 2022). In recent years, HF has been explored as a strong inhibitor of cytoplasmic ProRSs from *Plasmodium falciparum* (the causative agent of malaria) and *Toxoplasma gondii* (the agent of toxoplasmosis), both members of the E-type ProRS family. In contrast, HF is significantly less effective against P-type ProRS, such as *Pseudomonas aeruginosa* ProRS, which exhibits an IC$_{50}$ ~100-fold higher (Pena et al, 2019). Notably, the amino acid residues responsible for HF binding are highly conserved within the catalytic site of E-type ProRS, including TgProRS, but are poorly conserved in P-type ProRS (Ivanesthi et al, 2024; Jain et al, 2015), underscoring the structural basis for this selective inhibition. Beyond its antiparasitic applications, HF exhibits a broad range of pharmacological properties, including anti-fibrotic (Gnainsky et al, 2006), antiviral (Hwang et al, 2019), anti-inflammatory (Battu et al, 2018),

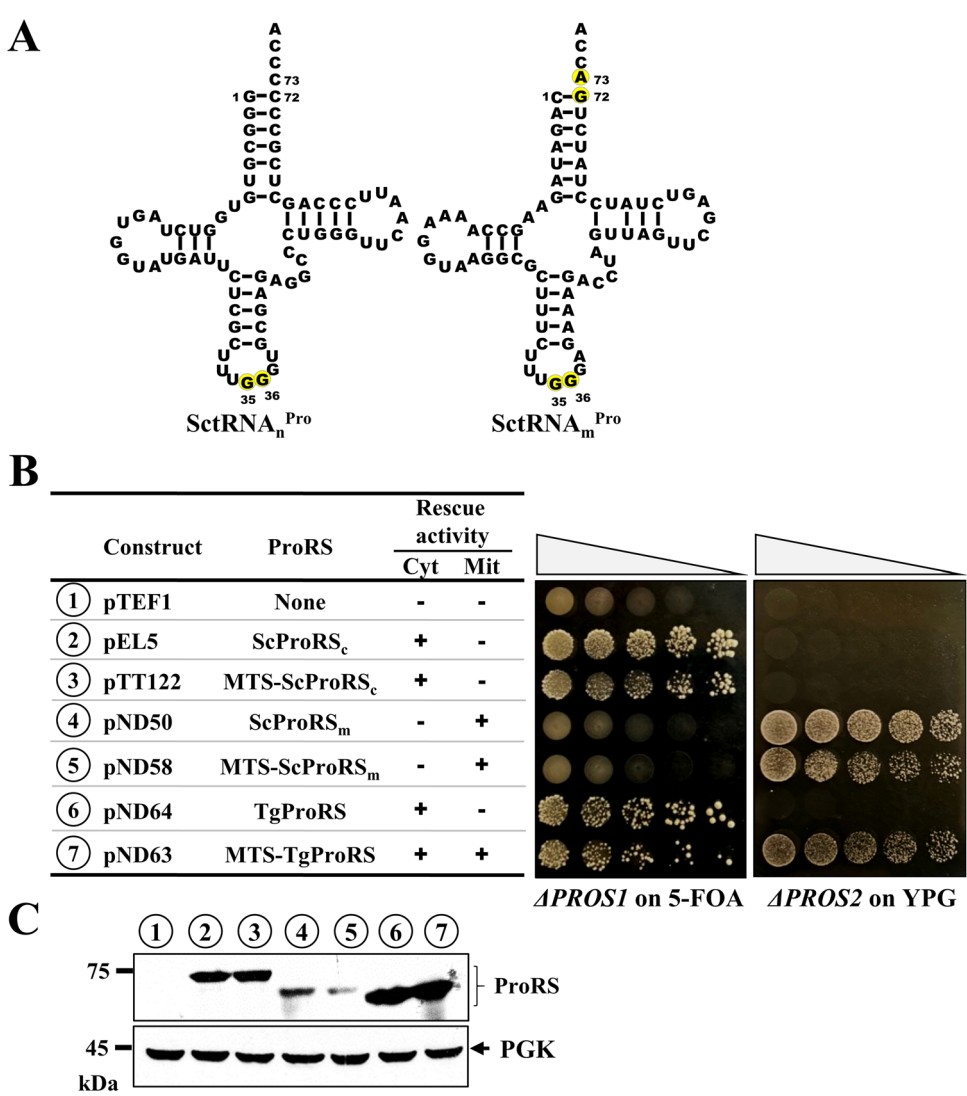

**Figure 3. Complementation assay of WT TgProRS.**

(A) Cloverleaf structure of tRNA$^{Pro}$. SctRNA$_n$$^{Pro}$, *S. cerevisiae* nuclear-encoded cytoplasmic tRNA$^{Pro}$; SctRNA$_m$$^{Pro}$, *S. cerevisiae* mitochondrial-encoded mitochondrial tRNA$^{Pro}$. (B) Summary of the ProRS constructs and their rescue activities. The symbols "+" and "−" respectively denote positive and negative complementation. Growth on 5-FOA and YPG, respectively, indicates complementation of the cytoplasmic and mitochondrial ProRS activities. (C) Western blotting. Expression of N-terminally His$_6$-tagged ProRSs from the plasmids was probed with an HRP-conjugated anti-His$_6$ tag antibody. Numbers 1–7 (circled) correspond to the constructs shown in (A). Source data are available online for this figure.

immunomodulatory (Wang et al, 2020), cardioprotective (Qin et al, 2017), and anticancer activities (Mi et al, 2022). The chemical structures of A2C and HF are shown in Fig. 5A.

Given that TgProRS charges both TgtRNA$_n$$^{Pro}$ and TgtRNA$_a$$^{Pro}$ with similar efficiency, we investigated whether it displays comparable IC$_{50}$ values for these tRNA substrates when exposed to the inhibitors A2C and HF. Inhibition assays with A2C revealed relatively similar IC$_{50}$ values for TgProRS: 1.3 mM for TgtRNA$_n$$^{Pro}$ and 1.6 mM for TgtRNA$_a$$^{Pro}$ (Fig. 5B)—likely because the proline-binding pocket adopts a similar conformation when accommodating either isoacceptor In contrast, HF inhibition assays uncovered a striking difference. TgProRS exhibited an IC$_{50}$ of 5 nM for TgtRNA$_n$$^{Pro}$, while the IC$_{50}$ for TgtRNA$_a$$^{Pro}$ was significantly higher at 24 nM—nearly a five-fold difference (Fig. 5C). This disparity

highlights the distinct geometric interactions between TgProRS and its tRNA substrates, leading to differences in HF binding affinity and inhibitory efficacy. Nonetheless, regardless of the tRNA$^{Pro}$ isoacceptor charged, the IC$_{50}$ values for TgProRS (5 nM for TgtRNA$_n$$^{Pro}$ and 24 nM for TgtRNA$_a$$^{Pro}$) remain substantially lower than those reported for bacterial E-type (~200 nM) and P-type (~1000 nM) ProRS enzymes (Ivanesthi et al, 2024; Latifah et al, 2024; Pena et al, 2019), highlighting its exceptional sensitivity to HF inhibition.

A possible explanation for the differences in IC$_{50}$ values could be variations in the affinities of TgtRNA$_n$$^{Pro}$ and TgtRNA$_a$$^{Pro}$ that are not captured in the $K_M$ values reported in Table 1, as these values represent the overall aaRS reaction rather than specific tRNA isoacceptor interactions. Another possible explanation is that

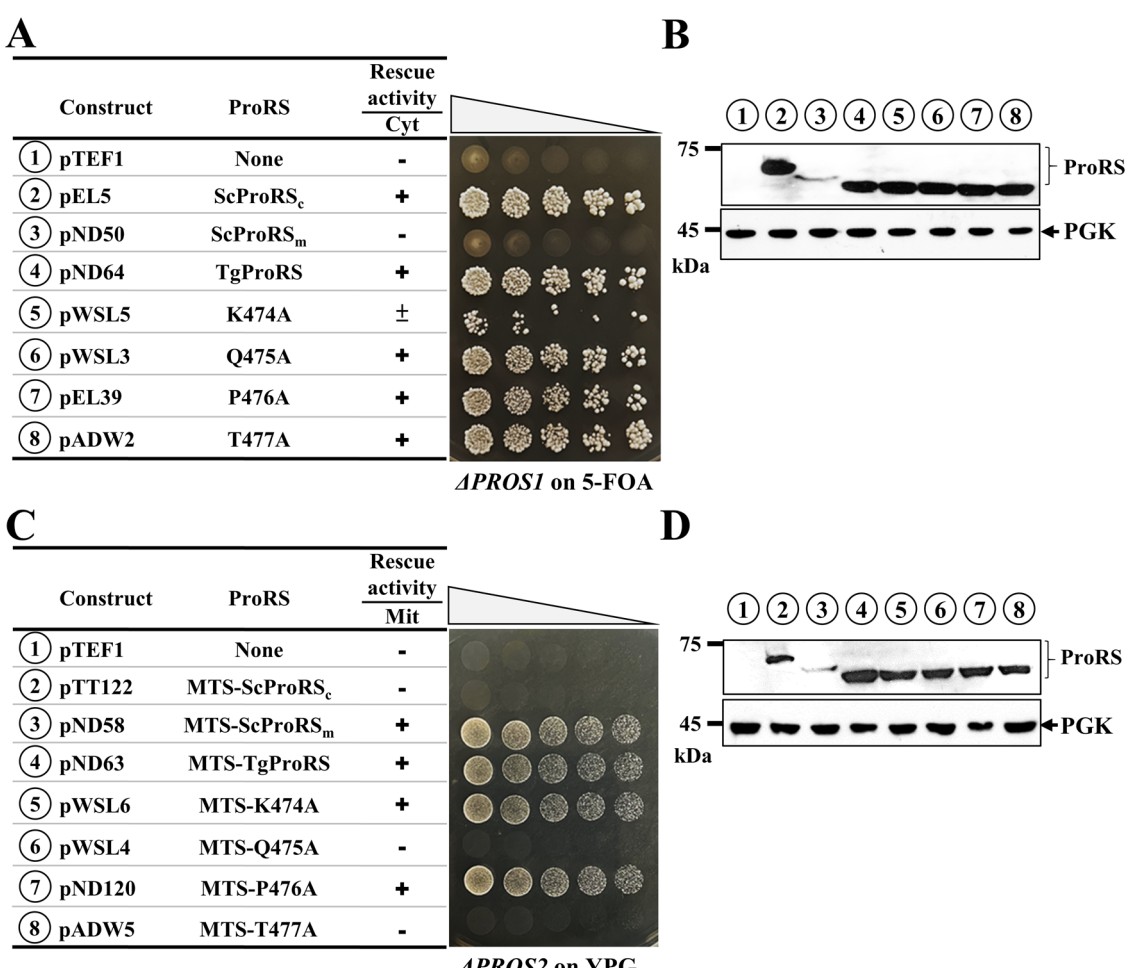

**Figure 4. Complementation assay of mutant TgProRSs.**

(A, C) Summary of the ProRS constructs and their rescue activities. The symbols "+" and "–", respectively, denote positive and negative complementation. Growth on 5-FOA and YPG, respectively, indicates complementation of the cytoplasmic and mitochondrial ProRS activities. (B, D) Western blotting. Expression of N-terminally His$_6$-tagged ProRSs from the plasmids was probed with an HRP-conjugated anti-His$_6$ tag antibody. Numbers 1–8 (circled) correspond to the constructs shown in (A, C). Source data are available online for this figure.

TgProRS undergoes conformational changes upon interacting with these two tRNA isoacceptors. A previous study demonstrated that the co-crystal structures of TgProRS bound to various HF derivative compounds reveal remarkable active-site plasticity, enabling the enzyme to accommodate different ligands (Jain et al, 2017). This structural adaptability suggests that TgProRS may also employ a similar degree of plasticity when interacting with different tRNA$^{Pro}$ isoacceptors. Such flexibility could enable the enzyme to undergo distinct adjustments depending on the specific tRNA substrates, resulting in differential IC$_{50}$ values in the HF inhibition assay.

Unfortunately, the available co-crystal structure of TgProRS:HF cannot provide insight into this possibility, as it does not include tRNA moiety, making it impossible to observe potential interactions or conformational changes that may occur upon tRNA binding. Additionally, the reported crystal structure of the TtProRS-tRNA$^{Pro}$ complex (PDB entry 1H4S) also fails to offer further clarity, as the 3′ end of the tRNA$^{Pro}$ acceptor stem does not extend into the active site (Yaremchuk et al, 2000), preventing a

complete understanding of how the enzyme interacts with the tRNA substrate. To fully elucidate the structural basis of TgProRS function and its potential conformational changes, it is crucial to determine the crystal structure of TgProRS in complex with its two tRNA$^{Pro}$ isoacceptors.

*T. gondii*, the cause of toxoplasmosis, poses significant health risks, especially to immunocompromised individuals and pregnant women. Its survival and virulence rely on its ability to adapt to diverse host environments (Manickam et al, 2022). TgProRS, a relaxed specificity ProRS, charges both E- and P-type tRNAs$^{Pro}$, enabling its function in the cytosol, mitochondrion, and apicoplast. These findings advance understanding of ProRS evolution and tRNA recognition while highlighting TgProRS as a promising target for antiparasitic drug development. Its unique ability to differentiate between tRNA$^{Pro}$ types provides a basis for developing more selective inhibitors with minimal host off-target effects, addressing challenges of resistance to existing therapies. However, it remains to be determined whether this relaxed specificity provides a survival advantage.

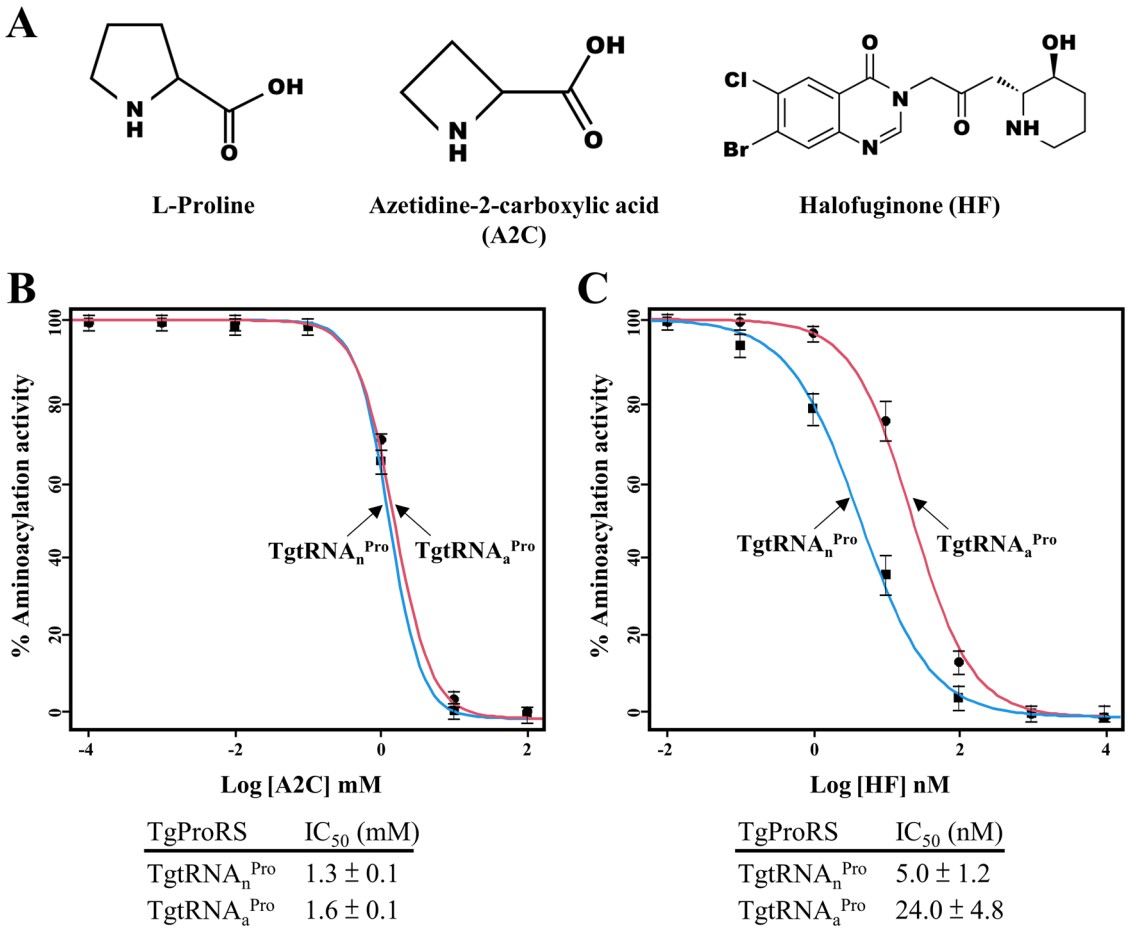

**Figure 5. Inhibition assay of TgProRS using A2C and HF.**

(A) Chemical structures of L-Proline, A2C, and HF. Aminoacylation assays were carried out at 37 °C for 10 min with varying (B) A2C (0.1 μM to 100 mM) and (C) HF concentrations (0.01 nM to 10 μM), respectively. Error bars represent mean ± SD ($n = 3$ technical replicates). Source data are available online for this figure.

| TgProRS | $IC_{50}$ (mM) |
|---|---|
| $TgtRNA_n^{Pro}$ | $1.3 \pm 0.1$ |
| $TgtRNA_a^{Pro}$ | $1.6 \pm 0.1$ |

| TgProRS | $IC_{50}$ (nM) |
|---|---|
| $TgtRNA_n^{Pro}$ | $5.0 \pm 1.2$ |
| $TgtRNA_a^{Pro}$ | $24.0 \pm 4.8$ |

## Methods

### Reagents and tools table

| Reagent/resource | Reference or source | Identifier or catalog number |
|---|---|---|
| **Experimental models** | | |
| DH10B (*E. coli*) | ThermoFisher | EC0113 |
| BL21-CodonPlus(DE3)-RIPL (*E. coli*) | Agilent Technologies | 230280 |
| INVSc1 (*S. cerevisiae*) | ThermoFisher | C81000 |
| *PROS1* KO strain (*S. cerevisiae*) | Ivanesthi et al, 2024 | N/A |
| *PROS2* KO strain (*S. cerevisiae*) | Ivanesthi et al, 2024 | N/A |
| **Recombinant DNA** | | |
| pQB169 | Cubist Pharmaceuticals, Inc., Lexington, MA | N/A |
| pET-19b | MERCK | 69677-M |

| Reagent/resource | Reference or source | Identifier or catalog number |
|---|---|---|
| Additional plasmids | This study | Dataset EV1 |
| **Antibodies** | | |
| Mouse Monoclonal 6X His tag® antibody- conjugated to HRP | Abcam | AD1.1.10 |
| Mouse Monoclonal anti-PGK1 antibody | Abcam | 22C5D8 |
| **Oligonucleotides and other sequence-based reagents** | | |
| PCR Primers | This study | Dataset EV2 |
| **Chemicals, enzymes, and other reagents** | | |
| NdeI | NEW ENGLAND Biolabs | Lot #10133989 |
| XhoI | NEW ENGLAND Biolabs | Lot #10161948 |
| T4 DNA Ligase | NEW ENGLAND Biolabs | M0202L |
| DpnI | NEW ENGLAND Biolabs | Lot #10196884 |

| Reagent/resource | Reference or source | Identifier or catalog number |
|---|---|---|
| Pfu polymerase | Promega | M7741 |
| Ribomax Large Scale RNA Production SYSTEM T7 | Promega | P1320 |
| MicroScint-O | Revvity | Part #: 6013611 |
| Proline, L-[2,3,4,5-³H]- | PerkinElmer, Waltham, MA, USA | Lot:2549211 |
| Whatman filters | GE Healthcare | Lot: 9790646 |
| Azetidine-2-carboxylic acid (A2C) | MERCK | A0760 |
| Halofuginone hydrobromide (HF) | MERCK | 32481 |
| L-Proline | MERCK | 1.07434 |
| Ni-NTA agarose | Qiagen | Cat no./ID. 30230 |
| Amersham ECL prime reagent | Cytiva | RPN2235 |
| **Software** | | |
| CLC Sequence Viewer version 8 | www.qiagenbioinformatics.com | |
| Quest Graph™ IC50 Calculator | https://www.aatbio.com/tools/ic50-calculator. | |
| R version 4.1.3 | The R Foundation for Statistical Computing | |
| **Other** | | |
| MicroBeta² | PerkinElmer | Product Code: 56145 |
| Cell disruptor | Beijer Electronic | |
| Semi-Dry Blotter | Invitrogen | SD1000 |
| X-ray film (Fuji Super RX-N) | Fujifilm | |

## Construction of plasmids

*T. gondii PROS* (Gene ID: TGME49_219850), lacking the sequence coding for OTS and ProXp-ala (amino acid residues 1–333) (Jain et al, 2017), was codon-optimized for *E. coli* expression (synthesized by *OMICS BIO*) with NdeI and XhoI as the cloning site, designated as pND55. After that, the NdeI/XhoI fragment was subcloned into pET-19b for protein purification, designated as pND67. For yeast rescue assays, the fragment was cloned into the NdeI/XhoI sites of pTEF1 with or without MTS (Ivanesthi et al, 2024). This plasmid contains the TEF1 promoter with an N-terminal His₆-tag and *LEU2* marker, derived from pQB169 (Cubist Pharmaceuticals, Inc., Lexington, MA). The fusion of a heterologous MTS to ProRS followed a previously described strategy, in which a DNA sequence encoding amino acid residues 1–46 of the mitochondrial precursor form of yeast valyl-tRNA synthetase was PCR-amplified as an XbaI-SpeI fragment and inserted into the 5' SpeI site of the ProRS gene (Ivanesthi et al, 2023). The TgProRS clone without MTS was designated as pND64, and the one with MTS was designated as pND63.

For creating *T. gondii PROS* or its motif 2 loop mutants, site-directed mutagenesis (SDM) was performed using pND55 as the template. The K474A mutation was introduced using primers EL10/EL11, Q475A using EL28/EL29, P476A using EL62/EL63, and T477A using EL30/EL31. The resulting mutant plasmids were named pEL12, pEL28, pEL37, and pEL23, respectively. The NdeI/XhoI fragment of TgProRS K474A from pEL12 was subcloned into pET-19b (designated as pEL13), pTEF1 (designated as pWSL5), and pTEF1 with MTS (designated as pWSL6). The NdeI/XhoI fragment of TgProRS Q475A from pEL28 was subcloned into pET-19b (designated as pEL29), pTEF1 (designated as pWSL3), and pTEF1 with MTS (designated as pWSL4). The NdeI/XhoI fragment of TgProRS P476A from pEL37 was subcloned into pET-19b (designated as pEL38), pTEF1 (designated as pEL39), and pTEF1 with MTS (designated as pND120). The NdeI/XhoI fragment of TgProRS T477A from pEL23 was subcloned into pET-19b (designated as pEL24), pTEF1 (designated as pADW2), and pTEF1 with MTS (designated as pADW5). *E. coli* DH10B strain was used for cloning and plasmid propagation.

## Protein purification

Plasmids harboring *T. gondii PROS* or its motif 2 loop mutants were individually transformed into *E. coli* BL21-CodonPlus(DE3)-RIPL and plated on LB medium containing 100 µg/ml ampicillin (LB/Amp). A single colony was grown overnight in 50 ml LB/Amp, then added to 1 L fresh LB/Amp and incubated at 30 °C until $OD_{600}$ reached 0.8–1.0. Protein expression was induced with 1 mM IPTG and shaken at 30 °C for 4 h. Cells were harvested by centrifugation, resuspended in 25 ml of Breaking Buffer with 10% glycerol, 50 mM $Na_2HPO_4$, 4 mM $NaH_2PO_4$, 300 mM NaCl, 0.1% Triton X-100, 2 mM PMSF and supplemented with 50 µl of protease inhibitor cocktail (Sigma-Aldrich). Cells were disrupted using a cell disruptor (Beijer Electronics), and debris was removed by centrifugation. The supernatant was mixed with equilibrated Ni-NTA resin (Qiagen) and incubated in 4 °C for 10 min with constant shaking, followed by centrifugation and supernatant removal. The resin was washed once with Wash Buffer I (10% glycerol, 6 mM $Na_2HPO_4$, 44 mM $NaH_2PO_4$, 300 mM NaCl, 5 mM 2-ME, 0.1% Triton X-100, 2 mM PMSF), twice with Wash Buffer II (10% glycerol, 6 mM $Na_2HPO_4$, 44 mM $NaH_2PO_4$, 1 M NaCl, 5 mM 2-ME, 0.1% Triton X-100, 2 mM PMSF and 80 mM Imidazole), and once more with Wash Buffer I, with gentle shaking for 5 min in a cold room before centrifugation and supernatant removal. The $His_{10}$-tagged protein was eluted with 10 ml of Elution Buffer (10% glycerol, 6 mM $Na_2HPO_4$, 44 mM $NaH_2PO_4$, 50 mM NaCl, 5 mM 2-ME, 2 mM PMSF, and 300 mM Imidazole) and the eluents were collected. Protein purity was assessed using 10% SDS-PAGE, and the preparation was concentrated to 1 ml using PEG4000 in 4 °C for 12 h. The protein was dialyzed Dialysis Buffer (10% glycerol, 20 mM Tris-HCl pH 7.4, 50 mM NaCl, 5 mM 2-ME) overnight, and Storage Buffer (40% glycerol, 20 mM Tris-HCl pH 7.4, 50 mM NaCl, 5 mM 2-ME) for 12 h, all at 4 °C. Finally, the purified protein was aliquoted (0.25 ml/tube) and stored at –80 °C.

## Rescue of the genetic loss of yeast *PROS1* on 5-FOA

In this study, yeast haploid knockout strains (*PROS1⁻*) were used to conduct functional complementation assays, following a previously

established strategy (Ivanesthi et al, 2024). To evaluate whether a heterologous ProRS could functionally replace the yeast cytoplasmic ProRS, the target gene was introduced into a *PROS1* knockout yeast strain. This strain also carried a maintenance plasmid containing the wild-type (WT) yeast *PROS1* gene and a *URA3* marker. Cultures were adjusted to an optical density ($A_{600}$) of 1.0, serially diluted in threefold steps, and 10-μl aliquots from each dilution were spotted onto 5-FOA-containing plates (1 mg/ml). The plates were incubated at 30 °C for 3–5 days. Since yeast cells expressing *URA3* convert 5-FOA into a toxic product, only transformants that successfully evict the maintenance plasmid could grow on these plates. Growth on 5-FOA plates confirmed that the introduced plasmid encoded a functional ProRS capable of charging yeast cytoplasmic tRNA^Pro, thereby supporting cell viability.

## Rescue of the genetic loss of yeast *PROS2* on YPG

For functional complementation assays, yeast haploid knockout strains (*PROS2⁻*) were utilized in this study (Ivanesthi et al, 2024). To determine whether a heterologous ProRS could effectively replace the yeast mitochondrial ProRS, the target gene was introduced into a *PROS2* knockout strain containing a maintenance plasmid carrying the wild-type (WT) yeast *PROS2* gene and a *URA3* marker. Transformants were plated on 5-FOA medium (1 mg/ml) to select for cells that had lost the maintenance plasmid. After 5-FOA selection, surviving transformants were spotted onto YPG plates and incubated at 30 °C for 3–5 days to evaluate growth. YPG medium, which uses glycerol as the sole carbon source, requires functional mitochondria for oxidative phosphorylation. Therefore, growth on the YPG plate confirmed that the test plasmid encoded a functional ProRS capable of charging mitochondrial tRNA^Pro, ensuring sufficient mitochondrial activity for cell survival.

## Protein extraction and western blotting

To evaluate the expression of the heterologous ProRSs, the target gene was introduced into an INVSc yeast strain and plated into SD/-Leu agar. A single yeast transformant was picked and inoculated into 4 ml of SD/-Leu broth, then incubated overnight at 30 °C with constant shaking (220 rpm). Cells were collected by centrifugation at 13,000 rpm for 1 min, and the supernatant was discarded. The pellet was resuspended in 100 μl of Lysis Buffer (containing 50 mM Tris-HCl, pH 7.4, 150 mM NaCl, 20 mM EDTA, 0.5% Triton X-100, 0.5% SDS, 10 mM PMSF) along with ~100 μl of beads, and incubated on ice for 2–3 min. Cells were lysed by vortexing for 3 min at 30-second intervals at 4 °C, followed by centrifugation at 13,000 rpm for 1 min to remove cell debris. The supernatants were supplemented with a 1X SDS-loading dye and heated to 95 °C before being loaded onto an SDS-PAGE gel. Following electrophoresis, proteins were transferred onto PVDF membranes (GE Healthcare) using the Semi-Dry Blotter (Invitrogen) in accordance with the manufacturer's guidelines. Immunoblotting was performed using anti-His-HRP (1:5000, Abcam) and anti-PGK-HRP (1:5000, Abcam) antibodies. Protein detection was carried out using the ECL Prime Western Blotting Detection Reagent (Cytiva). Image acquisition was conducted by exposing the blot to X-ray film (Fujifilm).

## Preparation of tRNA^Pro transcripts

In vitro transcription of tRNA^Pro was carried out according to a previously described protocol (Antika et al, 2022; Latifah et al, 2024). The transcription template was generated by PCR amplification of an insert containing a T7 promoter and the target tRNA gene. Plasmids used as templates for tRNA insert amplification are listed in Dataset EV1. The 5′ and 3′ primers used to generate mutant TgtRNA_n^Pro and TgtRNA_a^Pro are listed in Dataset EV2. In vitro transcription of tRNA^Pro was carried out using 0.3 μM T7 RNA polymerase at 37 °C for 3 h in a reaction buffer containing 20 mM Tris-HCl (pH 8.0), 150 mM NaCl, 20 mM MgCl_2, 5 mM DTT, 1 mM spermidine, and 2 mM of each NTP (Promega). The resulting tRNA^Pro transcript was purified via 10% polyacrylamide gel electrophoresis in 8 M urea. After ethanol precipitation and vacuum-drying, the tRNA pellet was dissolved in 1× TE buffer (20 mM Tris-HCl, pH 8.0, 1 mM EDTA) and refolded by heating to 80 °C, then slowly cooled to room temperature with 10 mM MgCl_2. Approximately 80% of the in vitro-transcribed tRNA^Pro was active in aminoacylation.

## Aminoacylation assay

Aminoacylation assay was carried out in a buffer containing 50 mM HEPES (pH 7.2), 0.2 mg/ml BSA, 25 mM KCl, 12 mM MgCl_2, 2 mM DTT, 2 mM 2-mercaptoethanol, 1 mM spermine, 4 mM ATP, 10 μM in vitro-transcribed tRNA^Pro, and 21.13 μM proline (1.09 μM ^3H-proline; PerkinElmer, Waltham, MA, USA) (Ivanesthi et al, 2024). To terminate the reactions, 10-μl aliquots of the reaction mixture were spotted onto Whatman filters (GE Healthcare) pre-soaked in 5% trichloroacetic acid (TCA) and 2 mM proline. The filters were then washed three times with ice-cold 5% TCA for 15 min each, dried, and submerged in scintillation liquid, MicroScint-O (Revvity), followed by liquid scintillation counting using the MicroBeta² (PerkinElmer). The data presented are averages from three independent experiments. Active protein concentrations were determined through active-site titration, as previously described (Fersht et al, 1975). The kinetic parameters for tRNA prolylation were determined by monitoring the initial charging rates within the first 2 min of the reaction. Aminoacylation assays were conducted at 37 °C using TgtRNA_n^Pro (and its mutant) and TgtRNA_a^Pro (and its mutant) at concentrations ranging from 1 to 32 μM, while TgProRS concentrations varied between 100 nM and 10 μM. Kinetic values were derived from Lineweaver–Burk plot analyses, with error margins representing standard deviations based on averages from three independent experiments (Chang et al, 2016; Lee et al, 2017).

## Inhibition assay

For the A2C and HF inhibition assays, A2C (MERCK) (ranging from 0.1 μM to 100 mM) and HF (MERCK) (ranging from 0.01 nM to 10 μM) were each individually mixed with the tested enzyme (at a final concentration of 100 nM) in a 50-μl aminoacylation buffer. The mixture was incubated at 37 °C for 10 min to allow for inhibition. Reactions were subsequently terminated as described above. IC_{50} values were calculated using Quest Graph™ IC50 Calculator (AAT bioquest, Inc. 2025).

## Data availability

This study includes no data deposited in external repositories.

The source data of this paper are collected in the following database record: biostudies:S-SCDT-10_1038-S44319-025-00457-x.

## Peer review information

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

## Acknowledgements

This work was supported by the National Science and Technology Council, Taiwan [NSTC113-2311-B-008-001 to CCW], and the Taipei Veterans General Hospital [VGHUST113-G4-1-1 to CCW]. Funding for open access charges was provided by the National Science and Technology Council, Taiwan [NSTC113-2311-B-008-001 to CCW and NSTC113-2118-M-008-003-MY2 to YKT].

## Author contributions

**Indira Rizqita Ivanesthi**: Conceptualization; Data curation; Software; Formal analysis; Validation; Investigation; Visualization; Methodology; Writing—original draft; Writing—review and editing. **Emi Latifah**: Conceptualization; Data curation; Software; Formal analysis; Validation; Investigation; Visualization; Methodology; Writing—review and editing. **Shih-Yang Liu**: Software; Formal analysis; Validation; Project administration; Writing—review and editing. **Yi-Kuan Tseng**: Resources; Formal analysis; Supervision; Funding acquisition; Methodology. **Hung-Chuan Pan**: Data curation; Validation; Investigation. **Chien-Chia Wang**: Conceptualization; Resources; Supervision; Funding acquisition; Validation; Methodology; Writing—review and editing.

Source data underlying figure panels in this paper may have individual authorship assigned. Where available, figure panel/source data authorship is listed in the following database record: biostudies:S-SCDT-10_1038-S44319-025-00457-x.

## Disclosure and competing interests statement

The authors declare no competing interests.

# Expanded View Figures

**Figure EV1. P-type tRNA^Pro isoacceptors from bacteria and eukaryote's organelles.**

Cloverleaf structure of tRNA^Pro. TgtRNA$_a$^Pro, *T. gondii* apicopast-encoded apicoplast tRNA^Pro; EctRNA^Pro, *E. coli* tRNA^Pro; BttRNA^Pro, *Bacillus thuringiensis* tRNA^Pro; EftRNA^Pro, *Enterococcus faecalis* tRNA^Pro; BstRNA^Pro, *Bacillus subtilis* tRNA^Pro; HstRNA$_m$^Pro, *Homo sapiens* mitochondrial-encoded mitochondrial tRNA^Pro; MmtRNA$_m$^Pro, *Mus musculus* mitochondrial-encoded mitochondrial tRNA^Pro; AttRNA$_m$^Pro, *Arabidopsis thaliana* mitochondrial-encoded mitochondrial tRNA^Pro; DmtRNA$_m$^Pro, *Drosophila melanogaster* mitochondrial-encoded mitochondrial tRNA^Pro.

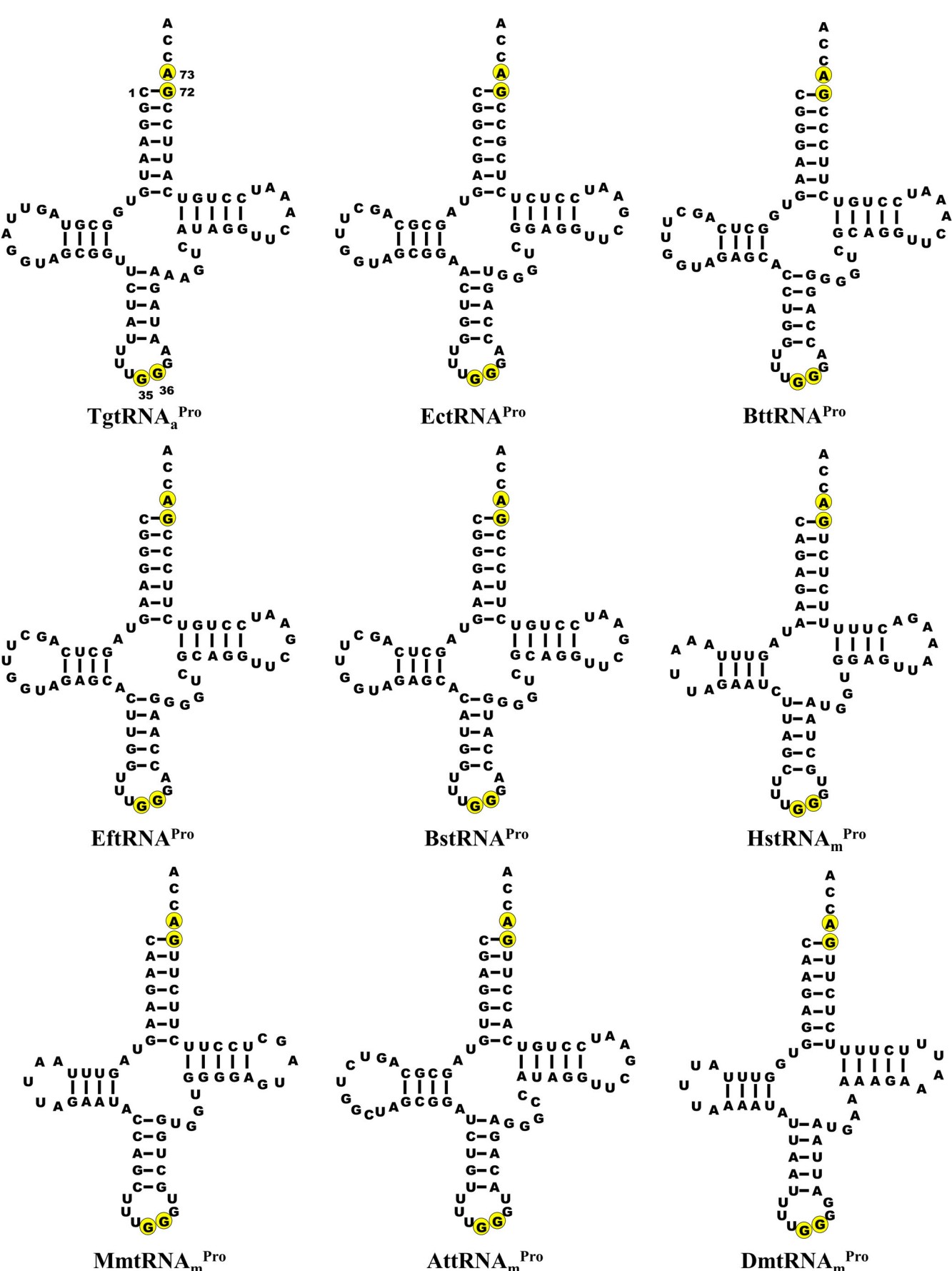

