## [Peer Review File · EMBO Reports]

Dual-mode recognition of tRNA^{Pro} isoacceptors by *Toxoplasma gondii* Prolyl-tRNA synthetase

Indira Rizqita Ivanesthi, Emi Latifah, Shih-Yang Liu, Yi-Kuan Tseng, Hung-Chuan Pan, and Chien-Chia Wang

Corresponding author(s): Chien-Chia Wang (dukewang@cc.ncu.edu.tw)

Review Timeline:

Submission Date:	11th Feb 25
Editorial Decision:	11th Mar 25
Revision Received:	25th Mar 25
Editorial Decision:	8th Apr 25
Revision Received:	11th Apr 25
Accepted:	15th Apr 25

Editor: Achim Breiling

Transaction Report:

Dear Prof. Wang,

Thank you for the transfer of your manuscript to EMBO reports. I have now received the reports from the three referees that were asked to evaluate your study, which can be found at the end of this email.

As you will see, the referees think that these findings are of interest. However, they have several comments, concerns, and suggestions, indicating that a major revision of the manuscript is necessary to allow publication of the study in EMBO reports. As the reports are below, and all the referee concerns need to be addressed, I will not detail them here.

Given the constructive referee comments, I would like to invite you to revise your manuscript with the understanding that the concerns of the referees must be addressed in the revised manuscript and in a detailed point-by-point response. Acceptance of your manuscript will depend on a positive outcome of a second round of review. It is EMBO reports policy to allow a single round of revision only and acceptance of the manuscript will therefore depend on the completeness of your responses included in the next, final version of the manuscript.

- 1) a .docx formatted version of the final manuscript text (including legends for main figures, EV figures and tables), but without the figures included. Figure legends should be compiled at the end of the manuscript text.
- 2) individual production quality figure files as .eps, .tif, .jpg (one file per figure), of main figures and EV figures. Please upload these as separate, individual files upon re-submission.

- 4) a complete author checklist, which you can download from our author guidelines (<https://www.embopress.org/page/journal/14693178/authorguide>). Please insert page numbers in the checklist to indicate where the requested information can be found in the manuscript. The completed author checklist will also be part of the RPF.

- 5) that primary datasets produced in this study (e.g. RNA-seq, ChIP-seq, structural and array data) are deposited in an

appropriate public database. If no primary datasets have been deposited, please also state this in a dedicated section (e.g. 'No primary datasets have been generated and deposited'), see below.

The accession numbers and database should be listed in a formal "Data Availability" section that follows the model below. This is now mandatory (like the COI statement). Please note that the Data Availability Section is restricted to new primary data that are part of this study. This section is mandatory. As indicated above, if no primary datasets have been deposited, please state this in this section

Data availability

8) Regarding data quantification and statistics, please make sure that the number "n" for how many independent experiments were performed, their nature (biological versus technical replicates), the bars and error bars (e.g. SEM, SD) and the test used to calculate p-values is indicated in the respective figure legends (also for EV and Appendix figures). Please also check that all the p-values are explained in the legend, and that these fit to those shown in the figure. Please provide statistical testing where applicable. Please avoid the phrase 'independent experiment', but clearly state if these were biological or technical replicates. Please also indicate (e.g. with n.s.) if testing was performed, but the differences are not significant. In case n=2, please show the data as separate datapoints without error bars and statistics. See also: <http://www.embopress.org/page/journal/14693178/authorguide#statisticalanalysis>

9) Please add scale bars of similar style and thickness to microscopic images, using clearly visible black or white bars (depending on the background). Please place these in the lower right corner of the images themselves. Please do not write on or near the bars in the image but define the size in the respective figure legend.

10) Please also note our reference format:

12) We now use CRedit to specify the contributions of each author in the journal submission system. CRedit replaces the author contribution section. Please use the free text box to provide more detailed descriptions and do NOT provide your final manuscript text file with an author contributions section. See also our guide to authors: <https://www.embopress.org/page/journal/14693178/authorguide#authorshipguidelines>

13) All Materials and Methods need to be described in the main text using our 'Structured Methods' format, which is required for

all research articles. According to this format, the Methods section should include a Reagents and Tools Table (listing key reagents, experimental models, software, and relevant equipment and including their sources and relevant identifiers), uploaded as separate file, and a Methods section in which we encourage the authors to describe their methods using a step-by-step protocol format with bullet points, to facilitate the adoption of the methodologies across labs. More information on how to adhere to this format as well as downloadable templates (.doc) for the Reagents and Tools Table can be found in our author guidelines (section 'Structured Methods'):

14) Please order the sections like this, using these names:

Title page - Abstract - Keywords - Introduction - Results - Discussion - Methods - Data availability section - Acknowledgements (including the funding information) - Disclosure and Competing Interests Statement - References - Figure legends - Expanded View Figure legends

15) Please make sure that all the funding information is also entered into the online submission system and that it is complete and similar to the one in the acknowledgement section of the manuscript text file.

16) Please remove the list of abbreviations from the manuscript text file. Please define each abbreviation upon first mention in the text.

Finally, please note that corresponding authors are required to supply an ORCID ID upon submission of a revised manuscript and an institutional e-mail address. Please find instructions on how to link the ORCID ID to the account in our manuscript tracking system in our Author guidelines: <http://www.embopress.org/page/journal/14693178/authorguide#authorshipguidelines>

I look forward to seeing a revised form of your manuscript when it is ready.

Yours sincerely,

Referee #1:

This study describes how the prolyl-tRNA synthetase (ProRS) of the human parasite *T. gondii* evolved a flexible substrate specificity to recognize proline tRNAs with distinct structural features encoded in the nucleus and apicoplast genomes. Using biochemical and cell-based assays, the authors demonstrated that the only encoded ProRS in *T. gondii* enables protein synthesis in three different cellular compartments due to its relaxed tRNA specificity. This represents the first example of a ProRS capable of recognizing P- and E-type tRNA^{Pro}. Moreover, mutational analysis revealed the key role of two residues in defining TgProRS's tRNA specificity.

The manuscript contains very interesting insights into the substrate specificity adaptation of TgProRS, which could be of significant interest for the field of aminoacyl-tRNA synthetases. However, the interpretation of the results needs to be better supported and contextualized with previous work. Similarly, the broad biological relevance is hindered by the specific focus on TgProRS. Other E-type ProRSs are known to carry the same residues that enable TgProRS to recognize two structural different tRNA^{Pro}, hinting that the relaxed specificity may not be unique to TgProRS. Other specific comments are outlined below for the authors' consideration.

- The term "relaxed specificity" may be more suitable than "dual-specificity" to describe *T. gondii* ProRS's tRNA specificity. The results presented show that *T. gondii* ProRS has specificity for tRNA^{Pro} to generate Pro-tRNA^{Pro}, regardless of the structural differences of the two tRNA^{Pro}. In my opinion, dual specificity suggests that TgProRS either uses an amino acid or tRNA other than proline or tRNA^{Pro}, or alternative, that TgProRS catalyzes a reaction different from tRNA aminoacylation.

- The conclusion that the G72/A73 is crucial for aminoacylation of apicoplast tRNA^{Pro} by *T. gondii* ProRS requires additional support. Considering the *T. gondii* tRNA^{Pro} variants tested, it is possible that introducing a mispaired base pair C1-C72 (in addition to the A73C) is more structurally disruptive than the G1-G72 base paired introduced in the nuclear tRNA variant.
- Is the *T. gondii* apicoplast tRNA^{Pro} defined as prokaryotic-type based on the A73/G72 bases? It is possible that this tRNA has evolved unique sequence/structural changes recognized by *T. gondii* ProRS. A more robust comparison with other P-type tRNA^{Pro} from bacteria and organelles should help identify co-evolution changes. Importantly, adding the structures of the *S. cerevisiae* cytosolic and mitochondrial tRNA^{Pro} will be helpful.
- The N-terminal extension domain of some yeast and other "lower eukaryotes" correspond to ProXp-ala (previously known as PrdX) (PMID: 14663147 and PMID: 33051185). Thus, the term "YbaK" should be corrected. Moreover, it is unclear why a truncated version of the *T. gondii* ProRS was used.
- The KQPT has been previously reported in *Drosophila melanogaster* ProRS (PMID: 11342535) and *Plasmodium falciparum* ProRS (PMID: 25047712), and crystal structures have revealed the KQPT sequence in *T. gondii* (PMID: 36854028).
- An *T. gondii* strain carrying the genomic mutation ProRS T477A was shown to be viable (PMID: 36854028). Can the authors reconcile this observation with the phenotypes observed in the yeast model (Fig 4)?
- The first sentence of the second paragraph on page 3 implies that E- and P-type ProRSs are only encoded with E- and P-type tRNA^{Pro}, respectively, and therefore they don't cross-acylate. This should be revised to clarify that E-type ProRSs exist in bacteria that encode P-type tRNA^{Pro} as shown previously.
- The suggestion that Tg ProRS undergoes distinct conformational changes when charging tRNA^{Pro} isoacceptors is not well-supported. Differences in aminoacylation inhibition by HF could be due to other factors such as its binding mode/site. Available crystal structures of TgProRS in complex with HF could offer better insights.
- The description of A2C as an ProRS inhibitor and substrate is confusing. Whether the previously observed growth inhibition caused by A2C is due to toxic misincorporation or ProRS inhibition should be clarified.
- Evolutionary adaptation for recognition of tRNAs for the same amino acid but with bacterial and eukaryotic identity elements also occurred for human lysyl-tRNA synthetase, which is encoded by a single nuclear gene and functions in the mitochondria and cytoplasm. The human LysRS recognizes mitochondrial and cytoplasmic tRNA^{Lys}, which carry distinct sequence and structural features (PMID: 9278442). Thus, TgProRS's relaxed tRNA specificity may not be unprecedented.
- It is unclear from the discussion how TgProRS's "dual specificity" could be key factor for survival advantage.

Minor suggestions:

- An explanation of what "class II aaRS" refers to will be helpful non-experts.
- Reference 6 should be replaced with the primary source that supports the statement (PMID: 11399074)

Referee #2:

In the paper entitled "Dual-mode recognition of tRNA^{Pro} isoacceptors by *Toxoplasma gondii* ProRS" by Prof. Chien-Chia Wang's group, the authors demonstrated that a single eukaryotic/archaeal (E-type) prolyl-tRNA synthetase (ProRS) of parasitic eukaryote *Toxoplasma gondii* (TgProRS) efficiently charges both prokaryotic (P-type (with G72/A73)) and E-type (with C72/C73) tRNA^{Pro} isoacceptors. They also succeeded functional compensation for yeast cytoplasmic and mitochondrial ProRS activities. Furthermore, they performed the mutational analysis in the motif 2 loop in TgProRS and inhibition experiments using azetidine (a proline mimic) and halofuginone (HF, a Pro-A76 mimic). Based on these results, they proposed the TgProRS's essential role in protein synthesis across the parasite's three major compartments: the cytosol, mitochondrion, and apicoplast. The authors' discovery is interesting and their approaches are scientifically sound. Prof. Chien-Chia Wang has been a specialist in this research field and I recommend the paper should be published in *EMBO reports* after proper revision.

Although the text is generally well-written, I am only concerned about the following. I would just like to suggest that the authors should develop a discussion that is a little more grounded in the results of the structural analysis, for the results of the inhibition experiments using HF, which caused the differences for TgtRNA_n^{Pro} and TgtRNA_a^{Pro}. In the current form, although they just describe "This finding suggests that TgProRS undergoes distinct conformational changes when charging different tRNA^{Pro} isoacceptors. Moreover, TgtRNA_n^{Pro} charging is more susceptible to HF inhibition than TgtRNA_a^{Pro} charging, likely because HF aligns more effectively with the active-site conformation of TgProRS when bound to the E-type tRNA^{Pro} isoacceptor." (Page 9-

10), this description is still vague. Because the complex structure of *T. thermophilus* ProRS and tRNA^{Pro} is available (Stephen Cusack's group), the recognition of not only HF but also of 72nd and 73rd positions of tRNA^{Pro}s should be somewhat discussed based on the structure. I wonder if they can do more effective discussion together with by comparing the sequence alignment shown in Fig. 1B.

Minor points:

*Page 2, Line 32

editing (4,5) → editing (Fig. 1A) (4,5)

*Page 2, Line 33

site (6) → site (Fig. 1A) (6)

*Page 3, Line 3

synthetase (7),(8) → synthetase (Fig. 1A) (7,8)

*Page 3, Line 3

loop (11) → loop (Shimizu *et al.* 1992 (see below), 11)

The paper by Shimizu *et al.* should also be cited:

Shimizu, M., Asahara, H., Tamura, K., Hasegawa, T. and Himeno, H. (1992) The role of anticodon bases and the discriminator nucleotide in the recognition of some *E. coli* tRNAs by their aminoacyl-tRNA synthetases. *J Mol Evol*, 35, 436-443.

*Page 7, Line 14

(22), HF → (22), Halofuginone (HF)

*Page 7, After Line 10, and Fig. 5

The chemical structures of azetidine-2-carboxylic acid (A2C) and halofuginone (HF) should be shown somewhere.

*Page 9, Line 13 and Line 27

tRNAs^{Pro} → tRNA_{Pro}s

Referee #3:

The manuscript by Ivanesthi *et al.* elegantly shows that *Toxoplasma gondii* ProRS, though classified as an E-type enzyme, can efficiently aminoacylate both cytosolic (E-type) and apicoplast (P-type) tRNA^{Pro} isoacceptors. Through comprehensive biochemical and yeast complementation assays, the authors demonstrate that TgProRS recognizes different identity elements in each tRNA isoacceptor and that specific motif 2 loop residues underlie this dual specificity. These findings challenge the long-held notion that ProRS enzymes cannot cross-acylate E- and P-type tRNAs. In addition to providing a better basic understanding, these findings may open new doors for antiparasitic drug development. However, a clear path for the translational strategy is not provided.

Overall, the study is well executed, with clear writing and a logical progression of results. My principal concerns center on the steady-state characterization and the halofuginone (HF) activity profile.

I am surprised by the low values of *k*_{cat} reported in this paper, which appear to be more than 3 orders of magnitude slower than comparable data, e.g, for *E. coli*. This seems to be surprising, yet no rationale is provided for this discrepancy.

Furthermore, while the authors attribute the reduced potency of HF against TgtRNA^{Pro} (relative to TgtRNA^{Pro}) to different conformations, it may equally stem from differential binding affinities that are not captured by *K_m* measurements.

Finally, a minor point concerns the use of commas versus periods in numerical data. References to IC₅₀ values such as "1,324 μM" likely intend "1.324 μM."

Manuscript Number: EMBOR-2025-61312-T

Article Title: Dual-mode recognition of tRNA^{Pro} isoacceptors by *Toxoplasma gondii* ProRS

Journal Name: *EMBO Reports*

Dear Editor,

We would like to thank you and the reviewers for your careful consideration of the above-referenced manuscript. We have carefully revised the manuscript as suggested by the reviewers. Responses to the reviewer's comments are given below.

Reviewer: 1

Major comments:

Q1: The term "relaxed specificity" may be more suitable than "dual-specificity" to describe *T. gondii* ProRS's tRNA specificity. The results presented show that *T. gondii* ProRS has specificity for tRNA^{Pro} to generate Pro-tRNA^{Pro}, regardless of the structural differences of the two tRNA^{Pro}. In my opinion, dual specificity suggests that TgProRS either uses an amino acid or tRNA other than proline or tRNA^{Pro}, or alternative, that TgProRS catalyzes a reaction different from tRNA aminoacylation.

A1: As suggested, we revised the term "dual-specificity" to "relaxed specificity". (page 4, line 7; page 5, line 26; page 6, line 32; page 8, line 8; page 9, lines 3 and 24; page 10, line 17; page 12, lines 13 and 19)

Q2: The conclusion that the G72/A73 is crucial for aminoacylation of apicoplast tRNA^{Pro} by *T. gondii* ProRS requires additional support. Considering the *T. gondii* tRNA^{Pro} variants tested, it is possible that introducing a mispaired base pair C1-C72 (in addition to the A73C) is more structurally disruptive than the G1-G72 base paired introduced in the nuclear tRNA variant.

A2: In the preparation of TgtRNA_a^{Pro}, C1 was deleted to achieve high yields of tRNA transcripts. Therefore, the loss of efficiency of G72/A73 mutations in TgtRNA_a^{Pro} was solely due to base mutations, not the disruption of the first base pair. Additionally, previous research has shown that deleting C1 from tRNA^{Pro} has minimal effects on aminoacylation. To further support our conclusion, we conducted additional kinetic assays. (Table 1) (page 5, lines 8-11, 16-23)

Q3: Is the *T. gondii* apicoplast tRNA^{Pro} defined as prokaryotic-type based on the A73/G72 bases? It is possible that this tRNA has evolved unique sequence/structural changes recognized by *T. gondii* ProRS. A more robust comparison with other P-type

tRNA^{Pro} from bacteria and organelles should help identify co-evolution changes. Importantly, adding the structures of the *S. cerevisiae* cytosolic and mitochondrial tRNA^{Pro} will be helpful.

A3: Yes, we classified TgtRNA_a^{Pro} as a P-type tRNA^{Pro} based on the criteria established in published studies (PMID: 11342535, PMID: 8127693), which define P-type tRNAs^{Pro} by the presence of highly conserved nucleotides G72 and A73 in the acceptor stem. Furthermore, after comparing the sequence and structure of TgtRNA_a^{Pro} with other P-type tRNA^{Pro} from bacteria and organelles, we found no significant differences among them, supporting our classification. As suggested, we included several P-type tRNA^{Pro} isoacceptors from both bacteria and organelles for comparison (Fig. EV1). Additionally, we incorporated the *S. cerevisiae* cytosolic and mitochondrial tRNA^{Pro} in Fig. 3A. (page 8, lines 15-18)

Q4: The N-terminal extension domain of some yeast and other "lower eukaryotes" correspond to ProXp-ala (previously known as PrdX) (PMID: 14663147 and PMID: 33051185). Thus, the term "YbaK" should be corrected. Moreover, it is unclear why a truncated version of the *T. gondii* ProRS was used.

A4: As suggested, we revised the term "YbaK" to "ProXp-ala" in Fig. 1A. (page 2, line 30; page 4, line 13; page 12, line 25).

We used the truncated version of TgProRS following the strategies from previously published papers (PMID: 28867614 and PMID: 36854028). In addition, a previous study (PMID: 17283340) shows that deleting the N-terminal domain ProXp-ala in ScProRS_c results in only a threefold reduction in k_{cat}/K_M compared to WT ScProRS_c, suggesting that the N-terminus plays a minimal role in aminoacylation.

Q5: The KQPT has been previously reported in *Drosophila melanogaster* ProRS (PMID: 11342535) and *Plasmodium falciparum* ProRS (PMID: 25047712), and crystal structures have revealed the KQPT sequence in *T. gondii* (PMID: 36854028).

A5: Yes, we also observed the presence of the KQPT motif in several other ProRS enzymes, including those from *Drosophila melanogaster* and *Plasmodium falciparum*. In both species, an additional P-type ProRS is responsible for charging tRNA^{Pro} in their organelles. It is also worth determining whether the cytoplasmic DmProRS_c and PfProRS_c exhibit the same relaxed specificity. (page 9, lines 28-33)

Q6: A *T. gondii* strain carrying the genomic mutation ProRS T477A was shown to be viable (PMID: 36854028). Can the authors reconcile this observation with the phenotypes observed in the yeast model (Fig 4)?

A6: In PMID: 36854028, it was reported that the plaque size of the T592S mutant was

slightly larger than that of the T477A mutant, suggesting that the T477A mutation leads to a slower growth rate compared to T592S. This indicates that T477A has some impact on *T. gondii* (page 9, lines 27-28). In our complementation assay using a yeast knockout model, this mutation in TgProRS also impaired its ability to support yeast growth on YPG, suggesting that the charging of yeast mitochondrial P-type tRNA^{Pro} is impaired—a scenario that aligns, at least partially, with the observation in *T. gondii*.

Q7: The first sentence of the second paragraph on page 3 implies that E- and P-type ProRSs are only encoded with E- and P-type tRNA^{Pro}, respectively, and therefore they don't cross-acylate. This should be revised to clarify that E-type ProRSs exist in bacteria that encode P-type tRNA^{Pro} as shown previously.

A7: As suggested, we revised the text accordingly. (page 3, lines 14-17)

Q8: The suggestion that TgProRS undergoes distinct conformational changes when charging tRNA^{Pro} isoacceptors is not well-supported. Differences in aminoacylation inhibition by HF could be due to other factors such as its binding mode/site. Available crystal structures of TgProRS in complex with HF could offer better insights.

A8: We revised the use of the term 'conformational changes' throughout the manuscript.

A possible explanation for the differences in IC₅₀ values could be variations in the affinities of TgtRNA_n^{Pro} and TgtRNA_a^{Pro} that are not captured in the K_M values reported in Table 1, as these values represent the overall aaRS reaction rather than specific tRNA isoacceptor interactions. Another possible explanation is that TgProRS undergoes conformational changes upon interacting with these two tRNA isoacceptors. A previous study (PMID: 28867614) demonstrated that the co-crystal structures of TgProRS bound to various HF derivative compounds reveal remarkable active-site plasticity, enabling the enzyme to accommodate different ligands. This structural adaptability suggests that TgProRS may also employ a similar degree of plasticity when interacting with different tRNA^{Pro} isoacceptors. Such flexibility could enable the enzyme to undergo distinct adjustments depending on the specific tRNA substrate resulting differential IC₅₀ in HF inhibition assay. (page 10, lines 29-33; 11, lines 1-7)

Q9: The description of A2C as a ProRS inhibitor and substrate is confusing. Whether the previously observed growth inhibition caused by A2C is due to toxic misincorporation or ProRS inhibition should be clarified.

A9: As suggested, we revised the text accordingly. (page 7, line 16-23)

Q10: Evolutionary adaptation for recognition of tRNAs for the same amino acid but

with bacterial and eukaryotic identity elements also occurred for human lysyl-tRNA synthetase, which is encoded by a single nuclear gene and functions in the mitochondria and cytoplasm. The human LysRS recognizes mitochondrial and cytoplasmic tRNA^{Lys}, which carry distinct sequence and structural features (PMID: 9278442). Thus, TgProRS's relaxed tRNA specificity may not be unprecedented.

A10: We agree with the reviewer's comment. While TgProRS is not the first aaRS to exhibit dual specificity, our study demonstrates that it is the first ProRS enzyme to do so. To clarify this point, we incorporated findings from PMID: 9278442 into our discussion. (page 8, lines 32-33; page 9, lines 1-5)

Q11: It is unclear from the discussion how TgProRS's "dual specificity" could be key factor for survival advantage.

A11: We agree with the reviewer's comment. The idea that TgProRS dual specificity could be a key factor in its survival advantage requires further investigation. Therefore, we revised our explanation in the discussion section. (page 12, lines 10-20)

Minor comments:

Q1: An explanation of what "class II aaRS" refers to will be helpful non-experts.

A1: As suggested, we included the explanation in our introduction. (page 2, line 17-21)

Q2: Reference 6 should be replaced with the primary source that supports the statement (PMID: 11399074).

A2: As suggested, we replaced it with the proper citation. (page 2, line 29)

Reviewer: 2

Major comments:

Q1: I would just like to suggest that the authors should develop a discussion that is a little more grounded in the results of the structural analysis, for the results of the inhibition experiments using HF, which caused the differences for TgtRNA_n^{Pro} and TgtRNA_a^{Pro}. In the current form, although they just describe "This finding suggests that TgProRS undergoes distinct conformational changes when charging different tRNA^{Pro} isoacceptors. Moreover, TgtRNA_n^{Pro} charging is more susceptible to HF inhibition than TgtRNA_a^{Pro} charging, likely because HF aligns more effectively with the active-site conformation of TgProRS when bound to the E-type tRNA^{Pro} isoacceptor." (Page 9-10), this description is still vague. Because the complex structure of *T. thermophilus* ProRS and tRNA^{Pro} is available (Stephen Cusack's group),

the recognition of not only HF but also of 72nd and 73rd positions of tRNA^{Pro} should be somewhat discussed based on the structure. I wonder if they can do more effective discussion together with by comparing the sequence alignment shown in Fig. 1B.

A1: A possible explanation for the differences in IC₅₀ values could be variations in the affinities of TgtRNA_n^{Pro} and TgtRNA_a^{Pro} that are not captured in the K_M values reported in **Table 1**, as these values represent the overall aaRS reaction rather than specific tRNA isoacceptor interactions. Another possible explanation is that TgProRS undergoes conformational changes upon interacting with these two tRNA isoacceptors. A previous study (PMID: 28867614) demonstrated that the co-crystal structures of TgProRS bound to various HF derivative compounds reveal remarkable active-site plasticity, enabling the enzyme to accommodate different ligands. This structural adaptability suggests that TgProRS may also employ a similar degree of plasticity when interacting with different tRNA^{Pro} isoacceptors. Such flexibility could enable the enzyme to undergo distinct adjustments depending on the specific tRNA substrate resulting differential IC₅₀ in HF inhibition assay. (page 10, lines 29-33; 11, lines 1-7)
The co-crystal structure of *T. thermophilus* ProRS and its tRNA^{Pro} was published by Stephen Cusack's group (PMID: 10970866). Unfortunately, in this co-crystal structure, the acceptor stem (including G72 and A73) is positioned outside the enzyme's catalytic site. As a result, there is no insight into how an E-type ProRS interacts with the acceptor stem of tRNA^{Pro} (whether P-type or E-type) (page 11, lines 8-17).

Minor comments:

Q1: *Page 2, Line 32

editing (4,5) → editing (Fig. 1A) (4,5)

A1: As suggested, we revised the text accordingly. (page 2, line 26)

Q2: Page 2, Line 33

site (6) → site (Fig. 1A) (6)

A2: As suggested, we revised the text accordingly. (page 2, line 29)

Q3: Page 3, Line 3

synthetase (7),(8) → synthetase (Fig. 1A) (7,8)

A3: As suggested, we revised the text accordingly. (page 2, line 31)

Q4: Page 3, Line 3

loop (11) → loop (Shimizu et al. 1992 (see below), (11). The paper by Shimizu et al. should also be cited: Shimizu, M., Asahara, H., Tamura, K., Hasegawa, T. and Himeno, H. (1992) The role of anticodon bases and the discriminator nucleotide in the recognition of some *E. coli* tRNAs by their aminoacyl-tRNA synthetases. *J Mol Evol*, 35, 436-443.

A4: As suggested, we revised the text accordingly. (page 3, line 7)

Q5: Page 7, Line 14

(22), HF → (22), Halofuginone (HF)

A5: As suggested, we revised the text accordingly. (page 7, line 23)

Q6: Page 7, After Line 10, and Fig. 5

The chemical structures of azetidine-2-carboxylic acid (A2C) and halofuginone (HF) should be shown somewhere.

A6: As suggested, we incorporate the chemical structures of A2C and HF in Fig 5A. (page 7, line 28)

Q7: Page 9, Line 13 and Line 27

tRNAs^{Pro} → tRNAPros

A7: We revised to tRNA^{Pro} isoacceptors (page 10, lines 3 and 18)

Reviewer: 3

Major comments:

Q1: I am surprised by the low values of k_{cat} reported in this paper, which appear to be more than 3 orders of magnitude slower than comparable data, e.g, for *E. coli*. This seems to be surprising, yet no rationale is provided for this discrepancy.

A1: The k_{cat} values of TgProRS are a bit lower than those reported for *E. coli* ProRS. However, the difference is much less than three orders of magnitude. The reported k_{cat} for *E. coli* ProRS is 0.27 s^{-1} (PMID: 7870582), while for TgProRS, it is 0.04 s^{-1} and 0.092 s^{-1} , representing only a 7-fold and 3-fold difference, respectively. These variations could be attributed to differences in experimental conditions, such as buffer composition or temperature.

Q2: Furthermore, while the authors attribute the reduced potency of HF against

TgtRNA_a^{Pro} (relative to TgtRNA_n^{Pro}) to different conformations, it may equally stem from differential binding affinities that are not captured by K_M measurements.

A2: We agree with the reviewer's comments. A possible explanation for the differences in IC₅₀ values could be variations in the affinities of TgtRNA_n^{Pro} and TgtRNA_a^{Pro} that are not captured in the K_M values reported in **Table 1**, as these values represent the overall aaRS reaction rather than specific tRNA isoacceptor interactions. Another possible explanation is that TgProRS undergoes conformational changes upon interacting with these two tRNA isoacceptors. A previous study (PMID: 28867614) demonstrated that the co-crystal structures of TgProRS bound to various HF derivative compounds reveal remarkable active-site plasticity, enabling the enzyme to accommodate different ligands. This structural adaptability suggests that TgProRS may also employ a similar degree of plasticity when interacting with different tRNA^{Pro} isoacceptors. Such flexibility could enable the enzyme to undergo distinct adjustments depending on the specific tRNA substrate resulting differential IC₅₀ in HF inhibition assay. (page 10, lines 29-33; 11, lines 1-7)

Minor comments:

Q1: Finally, a minor point concerns the use of commas versus periods in numerical data. References to IC₅₀ values such as "1,324 μM" likely intend "1.324 μM."

A1: As suggested, we have revised the numerical data to avoid any misunderstandings. Specifically, we have changed "μM" to "mM" in the A2C inhibition data (**Fig. 5B; page 9 line 33**)

Dear Prof. Wang,

Thank you for the submission of your revised manuscript to our editorial offices. I have now received the reports from the three referees that I asked to re-evaluate the study, you will find below. As you will see, the referees now fully support the publication of the study in EMBO reports. Referee #2 a minor point to correct.

Before we can proceed with formal acceptance, I have these editorial requests I ask you to address in a final revised manuscript:

- We plan to publish your manuscript as Report. For a Scientific Report we require that results and discussion sections are combined in a single chapter called "Results & Discussion". Please do this for your manuscript. For more details please refer to our guide to authors: <http://www.embopress.org/page/journal/14693178/authorguide#researcharticleguide>
- Please combine some figures to have not more than 5 final main figures. Then please update all the callouts.
- Please include the two tables in your main manuscript text with their legends (after the references - see below). Then please delete the separate file with the tables.
- Please order the manuscript sections like this, using these names:
Title page - Abstract - Keywords - Introduction - Results & Discussion - Methods - Data availability section - Acknowledgements - Disclosure and Competing Interests Statement - References - Tables - Figure legends - Expanded View Figure legends
- Please remove the sentence 'Source data are available online for this figure' from the legends. Source data will be directly linked to the respective figures.
- Please provide a complete author checklist, which you can download from our author guidelines (<https://www.embopress.org/page/journal/14693178/authorguide>). Please insert page numbers in the checklist to indicate where the requested information can be found in the manuscript and select answers for all the pull-down menus. The completed author checklist will also be part of the RPF.
- The uploaded Dataset EV2 has the wrong title in the Excel file (Dataset EV1). Please check.
- Please make sure that all the funding information is also entered into the online submission system and that it is complete and similar to the one in the acknowledgement section of the manuscript text file. Presently, the grant NSTC112-2311-B008-001 is missing in the submission system. Please check.
- Please use our reference format:
<http://www.embopress.org/page/journal/14693178/authorguide#referencesformat>

In addition, I would need from you uploaded separately:

I look forward to seeing a new revised version of your manuscript as soon as possible.

Best,

Referee #1:

The authors have addressed my comments.

Referee #2:

The authors have satisfactorily responded and made the necessary changes to the manuscript. It can be accepted, but only one point below should be fixed before publication.

Page 18, Line 1

"tRNAAla" should be "tRNA^{Ala}".

Referee #3:

All my concerns have been addressed adequately

All editorial and formatting issues were resolved by the authors.

Prof. Chien-Chia Wang
National Central University
Department of Life Sciences
300 Zhongda Rd. Zhongli District
Taoyuan 320317
Taiwan

Dear Prof. Wang,

I am very pleased to accept your manuscript for publication in the next available issue of EMBO reports. Thank you for your contribution to our journal.

Yours sincerely,
